# LEARNING OPEN-WORLD VISUAL-TACTILE GRASP STABILITY PREDICTION WITH SYNTHETIC DATA

## ABSTRACT

Grasp stability prediction with visual-tactile data is an important problem in robotics. Most prior work learns these predictors with limited real-world data. Moreover, their evaluation has also been restricted to a simple and unitary laboratory environment. Our work studies open-world visual-tactile grasp stability prediction, i.e. the predictor should zero-shot generalize to novel objects in novel environment. Towards this problem, we propose to learn with synthetic visual-tactile data, generated with FEM-based simulation and ray-tracing rendering. In our experiment, we show that our simulation pipeline has much higher physical fidelity, compared to the rigid-body simulation. Furthermore, the predictor trained on our synthetic dataset has higher accuracy on open-world grasp stability prediction tasks than models trained on real-world dataset or on synthetic dataset from rigid-body simulation.

## 1 INTRODUCTION

Robot grasping stability prediction is an important problem in robotics (Calandra et al., 2018; Zhang et al., 2023; Bekiroglu et al., 2016; Si & Yuan, 2022; Schill et al., 2012; Li et al., 2024b). For deployment safety in real-world robotics applications, it is important to have a predictor to evaluate whether the grip is firm and stable before moving the object around. However, most previous work (Kolamuri et al., 2021; Calandra et al., 2018; Li et al., 2024b) only trains and evaluates the predictor in a lab environment or with known objects (Si & Yuan, 2022; Zhang et al., 2023). This has limit the progress toward building zero-shot model that can be applied to any object, any robot in any environment. Therefore, we focus on learning open-world grasp stability predictors that can zero-shot generalize to novel objects and novel environments.

A common choice of previous work is to collect visual-tactile data from the real world with Gelsight sensors (Yuan et al., 2017) to train the model (Calandra et al., 2018; Zhang et al., 2023; Kolamuri et al., 2021). However, collecting visual-tactile data in real-world is cost-inefficient and thus hard to scale. In addition, current datasets are limited to a specific gripper, a fixed camera position, one type of tactile sensor, and the small set of objects. Clearly, scaling the real-world dataset for open-world prediction is not a feasible choice in the near future.

In our work, we propose learning open-world predictors with synthetic data that are generated with FEM-based simulation (Li et al., 2025b; 2020) and ray-tracing rendering (Section 3). To this end, we have developed a synthetic data generator with Taccel simulation (Li et al., 2025b) and IsaacSim rendering. Our physical grasping simulation is composed of three stages: finger closure stage, force adjustment stage, and gripper lifting stage, to simulate the stable and consistent grasping force in real-world grasping. Furthermore, we have also developed a tactile renderer that can produce high-quality depth maps and RGB tactile images. With our synthetic data generation pipeline, we have curated a large-scale visual-tactile grasping dataset for training. The dataset contains more than 30k visual-tactile data pairs from 10k unique grasps, generated with 453 objects. We have used GPT-4o (Achiam et al., 2023) and Paint3D (Zeng et al., 2024) in our dataset curation.

To evaluate predictors on open-world grasp stability prediction tasks, we collect a novel dataset (Section 4) with a hand-held UMI-based (Chi et al., 2024) gripper (Figure 3a). Here, we have collected an open-world dataset of 333 grasps and their outcomes for evaluation. In addition, we also collect a dataset with 5 3d-printed objects with grasp pose annotations. We use the grasp-annotated dataset to compare different simulation pipelines.

In the experiment, we first show that FEM-based simulation has a lower sim-to-real gap on soft-gel rigid-object interaction, compared to previous work Taxim (Si & Yuan, 2022) that uses rigid-body simulation, that is, PyBullet (Coumans & Bai, 2021). Moreover, it also results in more accurate tactile rendering that matches real-world tactile readings. Second, we show that training predictors with our synthetic dataset produces a predictor that achieves 77.5% average accuracy. This is significantly better than the model trained with a large existing real dataset (53%) and with a similar synthetic dataset generated with the Taxim simulation (61%). We also find that training with only visual or tactile data alone cannot produce a robust model. Thus, it is necessary to train visual-tactile models for grasp stability prediction.

In all, with our synthetic data generator, we open a new possibility to learn open-world grasp stability predictors that can be used on any gripper, any object in any environment.

## 2  RELATED WORKS

**Vision-based Tactile Sensors.** Vision-based tactile sensors, e.g. Gelsight (Yuan et al., 2017; Taylor et al., 2022), are the most widely used tactile sensors in robotics research. The tactile information is revealed by the gel's deformation and is captured by its built-in camera. Gelsight can capture fine-grained tactile information, as dense as fingerprints. In robotics, work has used vision-based sensors in many manipulation tasks, including grasping (Kolamuri et al., 2021; Calandra et al., 2018; Schill et al., 2012; Kolamuri et al., 2021; Bekiroglu et al., 2016; 2011; Zapata-Impata et al., 2019; Zhang et al., 2023; Li et al., 2024b), slip detection (Li et al., 2025a; James & Lepora, 2020; Li et al., 2018), deformable object manipulation (She et al., 2021; Wilson et al., 2023; Sunil et al., 2023), visual-tactile slam (Zhao et al., 2023; Suresh et al., 2024), contact-rich insertion (Dong et al., 2021; Kim & Rodriguez, 2022; Yu et al., 2023), and learning large visual-tactile models (Yang et al., 2024; Heng et al., 2025). Our work focuses on an important challenge in robotics, i.e. grasp stability prediction with both visual and tactile modalities. We design a synthetic data generator that generates visual-tactile training data for this problem.

**Grasp Stability Prediction.** Predicting whether the grip of the robot is stable is an important task in robotics (Zhao et al., 2024; Calandra et al., 2018; Schill et al., 2012; Kolamuri et al., 2021; Bekiroglu et al., 2016; 2011; Zapata-Impata et al., 2019). Many works use both vision and tactile / haptic information in the binary classification problem. They have targeted the problem with classic metrics (Bekiroglu et al., 2011), probabilistic models (Prattichizzo & Trinkle, 2016), and deep networks (Schill et al., 2012; Calandra et al., 2018). Most of the work collects real-world data to train the predictor but is limited in the scale and distribution of the dataset, as well as the scope of the evaluation. Some works (Si & Yuan, 2022; Zhang et al., 2023) have explored using synthetic visual-tactile data generated with rigid-body simulation. However, it has only been evaluated on known objects and in a lab environment. Our work studies learning grasp stability predictor that can generalize zero-shot to open-world novel objects and novel environment. To this end, we have developed a novel synthetic data generator based on FEM-based simulation, which we show has a smaller sim-to-real gap.

**Tactile Sensor Simulation.** Simulating accurate tactile sensor readings remains a challenge in robotics. The mechanics of common rigid body physics engines (Todorov et al., 2012; Makoviychuk et al., 2021) are different from real-world physics. To simulate vision-based tactile sensors, previous work has developed example-based simulators (Si & Yuan, 2022; Wang et al., 2022), differentiable simulators (Si et al., 2024), rigid body simulators coupled with penalty-based models (Akinola et al., 2025; Xu et al., 2023) and FEM-based simulators (Du et al., 2024; Chen et al., 2024; Li et al., 2025b). Si & Yuan (2022) has used an example-based simulator, built-in PyBullet (Coumans, 2015) for grasp stability prediction. However, our work shows that it has a large sim-to-real gap for objects with complex geometries. Alternatively, we choose the recent FEM-based simulation (Li et al., 2025b), built on the IPC engine (Li et al., 2020; Lan et al., 2022) for the generation of synthetic grasping data.

## 3  SYNTHETIC DATA GENERATION

Previous work (Si & Yuan, 2022) on visual-tactile synthetic data generation leverages rigid-body simulation (Coumans & Bai, 2021) and example-based tactile simulation. However, we find that

rigid body simulators cannot accurately model soft-gel and rigid-object contact dynamics and have a larger sim-to-real gap on objects with complex geometries (Section 5.1). Thus, alternatively, we use FEM-based simulation (Li et al., 2025b) built on the IPC framework (Li et al., 2020) (Section 3.1). Moreover, we use ray-tracing rendering built-on IsaacSim for photorealistic visual data and we leverage LLM (Achiam et al., 2023) to curate the grasping object dataset (Section 3.2).

### 3.1 FEM-BASED SIMULATION

Our generator uses Taccel (Li et al., 2025b) for physical simulation. Taccel is a recent simulation platform in robotics with IPC-based simulation (Li et al., 2020; Lan et al., 2022). It features memory-efficient computation, computation-efficient tactile-rendering, and user-friendly APIs.

Similarly to previous work (Zhang et al., 2023; Si & Yuan, 2022; Calandra et al., 2018), we focus on the setting of parallel grippers with two Gelsight Mini sensors (Figure 3a). Consequently, in simulation, we simplify the gripper with two $2.5\text{cm} \times 2\text{cm} \times 0.4\text{cm}$ tetrahedral FEM objects. We use these two FEM objects to simulate the gel of the Gelsight sensors. Moreover, the vertices of the tetrahedron are separated into two categories (Figure 1). The vertices on the back of the sensor are active nodes that are driven by the target position / velocity with kinematic stiffness as $5 \times 10^4$ during the simulation. All other vertices are passive nodes without explicit motion targets. For the tetrahedral FEM gels, we choose Young's modulus $E = 10^6\text{Pa}$ and Poisson ratio $\eta = 0.3$.

In our simulation, we load the grasping object as a rigid ABD object (Lan et al., 2022), together with its mass and friction coefficient. At initialization, the object is placed on flat ground under a stable pose, and the two-gel gripper is set at the sampled grasp pose. Then, we simulate the grasping process with the following 3 stages (Figure 1).

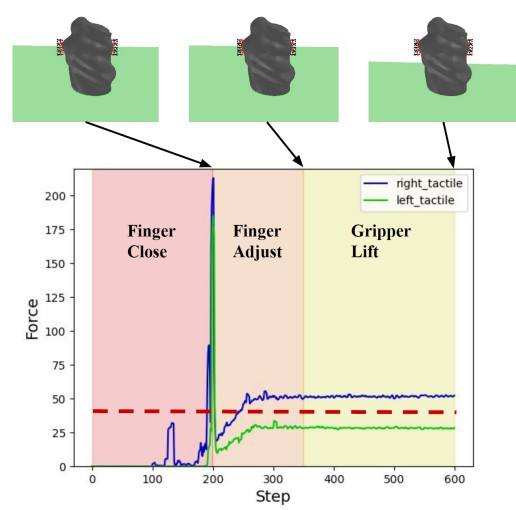

Figure 1: Contact force of both FEM gels (left and right fingers) during grasping process. We plot the forces of a stable grasp of the ORG object. The red dotted line shows $F_{\text{stop}} = 40$. We also show the physical state of the gels and the object at the end of each stage. Here, we use Trimesh to visualize the state. It is composed of the green ground, the gripping object and two set of point-clouds. The red points refer to the passive vertices of the gel and black points are the active points driven by target velocity. The images are rendered from the same pose relative to the gels in all three states.

1. **Finger Closure Stage.** We set the target velocity $v_c = 1\text{cm/s}$ to close the fingers (gels) along the closing direction. For each step, we compute the total contact force $f_c$ of all passive vertices, along their closing direction. When both gels' $f_c$ exceed a stopping threshold $F_{\text{stop}}$ or the fingers are fully closed, the simulation enters the force adjustment stage.

2. **Force Adjustment Stage.** To generate a stable grasping force on both gels, we use a simple closed-loop strategy to adjust the forces before lifting the object. Here, we set the target velocity as $v_a = -0.2\text{cm/s}$ if the average of both gel forces $\bar{f}_c > F_{\text{stop}}$ to open the gripper. Otherwise, we set $v_a = 0.2\text{cm/s}$ to close it. This stage lasts 1.5 seconds before the simulation enters the final gripper lifting stage.

3. **Gripper Lifting Stage.** The gripper (both gels) is lifted at a target velocity of $v_l = 2\text{cm/s}$ along the z-axis. Furthermore, we also apply a similar force adjustment strategy to stabilize the contact force $f_c$s during lifting. The simulation ends when the gripper reaches 5cm above its initial pose.

Finally, the grasp is considered a *success* if all points of the gripping object is above the ground more than 2cm. In our work, we render visual-tactile data of the last frame in the Force Adjustment Stage, and use it as the input data in the grasp stability prediction problem.

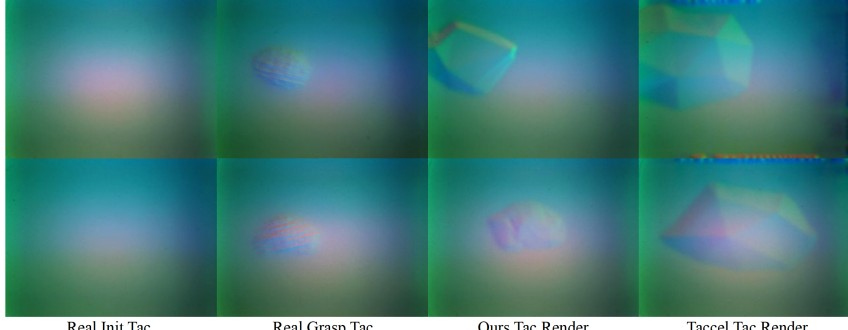

Real Init Tac     Real Grasp Tac     Ours Tac Render     Taccel Tac Render

Figure 2: Comparing ours tactile renderer and the built-in renderer with an example of the ORG object. The top row shows the left finger sensor and the bottom row shows the right finger sensor. From left to right: the real-world reading at initialization (no contact), the real-world reading before lifting (contact with ORG), the rendered simulation tactile reading with depth map from our method and OpenGL rendering, and the reading with the built-in method for depth map and OpenGL rendering for RGB.

To generate a high-quality simulation, we choose a small time step size of $\Delta t = 0.01$s, contact stiffness of $\kappa = 3 \times 10^6$kg $\cdot$ s$^{-2}$, contact distance threshold of $\hat{d} = 5 \times 10^{-4}$m, and friction transition of $\epsilon_v = 10^{-3}$m/s. Furthermore, we set a relatively tight tolerance $\epsilon_r = 5 \times 10^{-5}$ for the conjugate gradient (CG) solver and a maximum of $10^4$ CG steps. Although tight constraints result in a slow simulation speed, we find that this produces a more consistent contact force $f_c$ during the simulation process.

Figure 1 visualizes the contact force $f_c$ of both gels in a successful stable grasp of the ORG object (Figure 3d). We demonstrate the contact forces in each stage and the transition between stages w.r.t the forces. We notice that during the early contact of soft gel and the object in the Finger Closure stage, simulated contact forces are unstable. Therefore, we bring in the Force Adjustment Stage for a stable grasping force before lifting the object.

### 3.2 VISUAL-TACTILE DATASET GENERATION

**Objects Preparation.** We leverage objects and grasp poses from the ACRONYM dateset (Eppner et al., 2020). However, ACRONYM assets lack texture and an appropriate mass and friction coefficient. Thus, we use GPT-4o (Achiam et al., 2023) and a text-conditioned texture generation model Paint3D (Zeng et al., 2024) to estimate.

To be specific, with the object's category, we prompt GPT-4o for an appropriate texture type, then we use the Paint3D to generate the UV map for the mesh, conditioned on the texture type. In addition, we prompt GPT-4o to estimate the object mass and friction coefficient. In the LLM prompt, we include its category, dimension, and its texture type. Please refer to the Appendix for details of the prompt and preparation.

**Visual Rendering.** We use a ray-tracing rendering engine to render photorealistic visual RGB data, implemented with IsaacSim. With the textured mesh of the object, its pose, and poses of two gels from the simulation, we recompose the scene with IsaacSim Replicator. Here, we add the full gripper geometry, whose pose is computed on the basis of the gel poses. To generate diverse environment and lightning conditions, we add random ground texture and domelight lights with random HDRIs under random lumination intensity. The files are collected from PolyHaven [1]. Finally, we add the eye-in-hand camera at a similar location as the real gripper (Figure 3a), with additional randomization of its pose and focal length.

**Tactile Rendering.** We find that the built-in tactile renderer in Taccel (Li et al., 2025b) does not provide realistic tactile images. Taccel first computes the depth map by casting parallel rays from the image plane for each pixel. It identifies intersection points with the tetrahedron's surface and

---

[1]https://polyhaven.com

then uses a neural-network based renderer to render the RGB image given the depth map. However, the depth map has a large sim-to-real gap when we simulate the gel with relatively large tetrahedrons. Consequently, the rendered tactile image has a checkered pattern (Figure 2) and has a larger deformed area than the actual contact area. In contrast, tactile gels from the real world will deform more delicately (Figure 2).

Inspired by Li et al. (2024a), we implement our own tactile renderer with a simple modification of the built-in renderer. Here, we compute the intersection between the rays and the gripping object (not the gel surface), as we know the object mesh, its pose and the pose of the image plane where rays are casted. Then, we convert the intersection points into the depth map, clipped based on the thickness of the gel. The RGB image is then rendered with OpenGL similar to Li et al. (2024a). The colors and locations of the LED lights in the OpenGL renderer are carefully tuned to minimize the sim-to-real gap. Figure 2 shows the tactile rendering results of our tactile renderer, the rendered image with depth map from the Taccel's built-in renderer, and the ground truth. Clearly, ours renderer provides a more realistic tactile reading.

**Dataset Curation.** To generate our synthetic dataset, we use a total of 453 objects from the ACRONYM dataset. We simulate a total of 10k+ collision-free grasps from the ACRONYM under both $F_{\text{stop}} = 10, 40$N stopping thresholds. Approximately, it took 2500+ GPU hours for the physical simulation. However, we find that the dataset is imbalanced towards positive grasps and towards certain categories of objects, e.g. Mugs. To create a balanced data set for training, we sample grasps in categories that are disproportionate to others with a maximum number of grasps per category. In addition, we also sample positive grasps to create a training dataset balancing positive and negative grasps. Then, for each grasp, we render 3 RGB images with random background textures and lightnings, which took approximately 20+ GPU hours. In all, this creates a total of 30k+ visual-tactile data pairs of 10k+ unique grasps. Figure 5 shows examples of our dataset. Please refer to the Appendix for more examples.

## 4 REAL EVALUATION DATASET

To the best of our knowledge, there does not exist any real-world dataset that contains open-world visual-tactile data for grasp stability prediction. Thus, in our work, we collect a real-world evaluation dataset for our experiments (Section 5).

In order to collect open-world data easily, we design a hand-held gripper (Figure 3a) and use it to collect visual-tactile grasping data from the real world. The mechanical structure of the gripper is based on the UMI gripper (Chi et al., 2024). Here, we modify the finger design to attach two Gelsight Mini sensors and add a camera holder for a RealSense D435 camera. Noticing that we place the RGB camera of D435 at the center of the gripper for a symmetric RGB image. As the gripper is actuated by humans, it allows users to easily collect data in all kinds of environment and with different gripping forces. In our experiment, we collect two datasets for evaluation.

**Open-world Dataset.** We collect 60 objects from everyday life, some of them are from the YCB (Calli et al., 2015) object set. We also use transparent and reflective objects, such as glasses and forks. We place objects on flat tabletop surfaces in 10 different scenes, including classrooms, kitchens, and offices. Then, users randomly choose a grip pose for the gripper and actuate the grip. Before lifting the object, the user presses the keyboard to save the visual and tactile data at the current frame. Finally, he lifts the object and evaluates the grasp. In addition, we save the initial frame before closing the gripper. In total, we collect 333 grasps, including 163 negative grasps and 170 positive grasps. The open-world data set is used for the zero-shot evaluation of the grasp stability prediction models (Section 5.2). Here, we randomly split the dataset with 67 data for validation and 266 data for testing. Figure 3c illustrates the data collection process and Figure 3e shows the set of objects used in our dataset. Figure 5 shows examples of our open-world datasets. Please refer to the Appendix for more examples.

**Grasp-annotated Dataset.** In this dataset, we collect data on 5 3d-printed objects with complex geometries. Some of them are collected from the EGAD dataset (Morrison et al., 2020). These objects are usually considered adversarial objects for grasping (Mahler et al., 2017). We collect visual-tactile data similar to the open-world dataset.

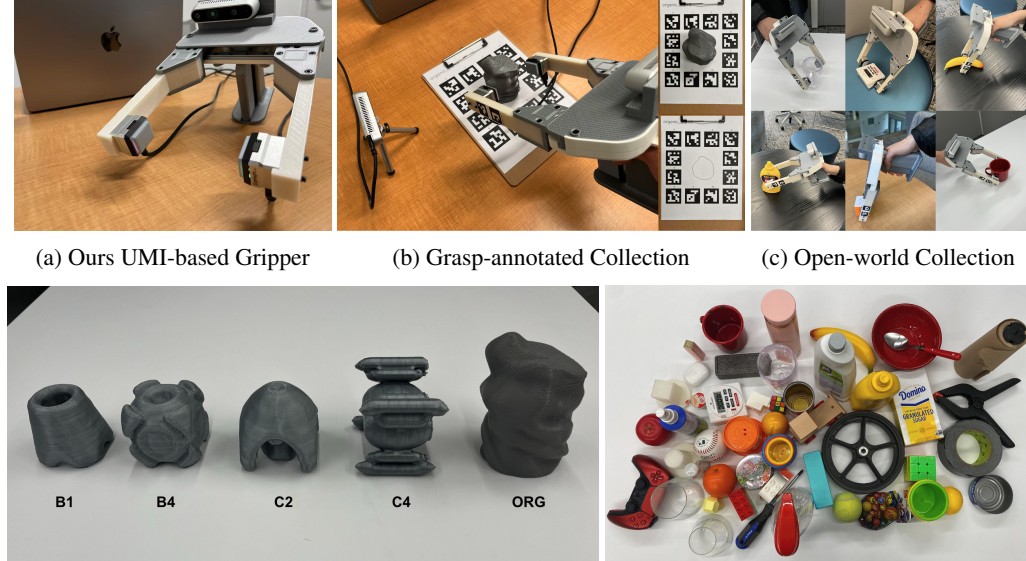

(a) Ours UMI-based Gripper (b) Grasp-annotated Collection (c) Open-world Collection

(d) Objects in Grasp-annotated Dataset (e) Objects in Open-world Dataset

Figure 3: (a) Illustration of the UMI-based (Chi et al., 2024) hand-held gripper for open-world data collection. (b) Illustration of data collection setup of our grasp-annotated dataset. The right two small figures illustrate the ArUco marker board we created, to align the object to its origin. (c) Illustration of data collection process of our open-world dataset. (d) The 5 3d-printed objects used in the Grasp-annotated Dataset. These objects have unusual geometry. From left to right, we name them as B1, B4, C2, C4, and ORG. (e) The set of everyday objects used in our open-world dataset, which contains transparent and reflective objects.

In addition, we also annotate the ground-truth grasp poses. This is achieved with ArUco markers and an external camera. We first placed the object on the ArUco board to let it align with the contour line of the object's bottom (Figure 3b). Then, we capture the relative grasp pose from the external RealSense camera, right before the gripper is actuated. The relative pose is calculated on the basis of ArUco markers on the board and on the gripper. Here, we collect 20 grasps for each of 5 objects. The grasp-annotated dataset is used to compare different simulation pipelines in terms of sim-to-real gaps. Please refer to the Appendix for data collection details.

## 5 EXPERIMENT

In our experiment, we first show that our simulation pipeline has a sim-to-real gap lower than that of rigid body simulation, on objects with complex geometries (Section 5.1). Then we show that data generated with our pipeline (Section 3) produce better models for the open-world grasp stability prediction problem, compared to using existing real-world dataset or with synthetic data generated from rigid body simulation (Section 5.2).

### 5.1 SIMULATION COMPARISON

Here, we show that our FEM-based simulation pipeline has a lower sim-to-real gap than rigid-body simulators, i.e. PyBullet used in Taxim (Si & Yuan, 2022). We replay all 20 grasps in the grasp-annotated dataset with our simulation (Section 3.1) and with Taxim. As we do not know the friction coefficient nor the grasping forces, we adopt the same experiment setup as in Si & Yuan (2022). We perform a grid search for the friction coefficient $\mu_o$ and gripping forces ($F_{\text{stop}}$) of each object. For each object, we report the highest accuracy of all combos that match the ground truth grasping outcomes. For fair comparison, we perform a similar sweep on the object's friction coefficient and joints' driving forces in the Taxim simulation. Please refer to the Appendix for details of the Taxim simulation and the best parameters.

Figure 4: Visualization of visual rendering and tactile rendering of the our simulation (second row) and with the Taxim (third row). We compare the results with the real-world ground truth of the same grasp (first row) from the ORG and the B4.

Table 1 compares accuracy between ours simulation and the Taxim. Clearly, ours has a lower sim-to-real gap, while the Taxim struggles on objects like B1, C4, and ORG. We believe that two factors contribute to the failure of Taxim. (1) rigid-body simulators like PyBullet require using convex decomposition for collision detection of the object. For objects that have complex and unusual geometries, convex decomposition will inevitably create artifacts for the contact. (2) The rigid-rigid contact in PyBullet is different from that in the real world of soft-gel and rigid object contact. The delicate deformation of the gel will result in more contact surfaces that wrap around the object. However, rigid-body simulation will usually result in point contact between the flat fingertip and the object.

In contrast, with FEM-based simulation (Li et al., 2020; Lan et al., 2022), we can not only avoid convex decomposition artifacts but also model the delicate soft-gel rigid-object contact (Kim et al., 2022) more accurately. Consequently, our physical simulation is more accurate in grasping outcomes.

Table 1: The accuracy of grasp outcome from the simulator that matches the real-world outcomes. We evaluate 20 grasps of each object on both simulators. Ours simulation produces quite accurate simulation, while the Taxim fails on objects like B1, C4, and ORG.

| Method (Simulator) | B1 | B4 | C2 | C4 | ORG | Avg |
|---|---|---|---|---|---|---|
| Taxim (Pybullet) | 0.55 | 0.85 | **0.95** | 0.65 | 0.65 | 0.73 |
| Ours (Taccel) | **0.9** | **1.0** | **0.95** | **0.9** | **0.95** | **0.94** |

Furthermore, we qualitatively compare the rendered tactile images between ours and the Taxim. Here, we use the same contact configuration of Taxim's tactile rendering as in its paper (Si & Yuan, 2022), but we change the background and lights' locations to match that of ours for Gelsight Mini sensors. Figure 4 shows the result of the visual-tactile data generated, together with the real-world ground truth of the same grasp. Clearly, we observe that our FEM-based simulator captures the delicate contact between the soft gel and the rigid object, as a similar tactile pattern can be observed in Figure 4. However, the tactile images rendered with Taxim quite different from the real world. It either misses a lot of information (left, the ORG) or creates many artifacts in the reading (right, the B4). This shows that our simulation has smaller sim-to-real gap in tactile rendering, owing to more accurate modeling of contact.

## 5.2 MODEL EVALUATION

We show that synthetic data generated with our method are better training data for open-world grasp stability prediction problem. Here, we compare with the following baseline datasets and Figure 5

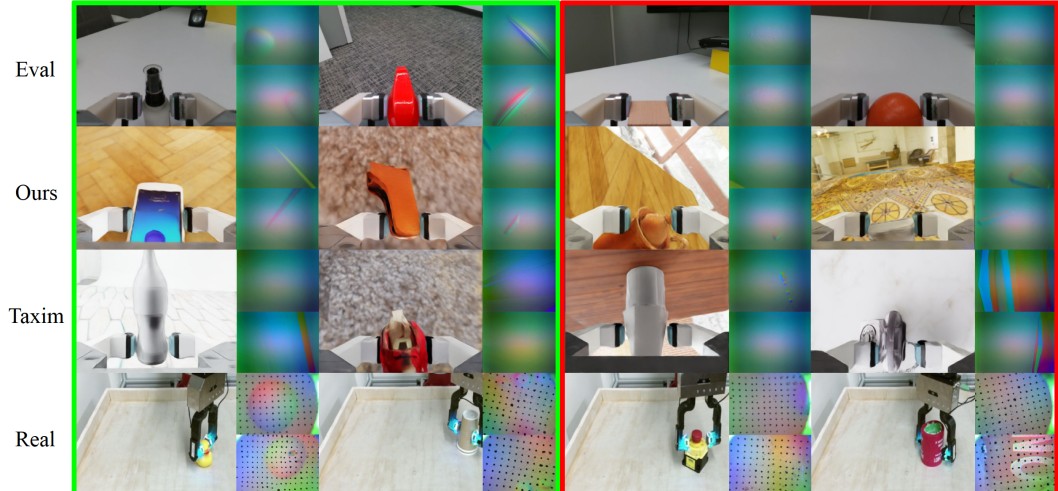

Figure 5: Examples of different visual-tactile dataset. The left two columns contain positive grasps and right two columns contain negative grasps. From top to the bottom, we show the our evaluation dataset, our synthetic training dataset, synthetic dataset genrated with Taxim, and the Real training dataset. For each case, we show the RGB image, the left tactile reading (top) and right tactile reading (bottom).

shows the examples of our real evaluation data, our synthetic training data, real-world training data, and sythetic data with the Taxim.

- **Real Dataset (Real)** (Calandra et al., 2018). The real-world dataset contains 9.2k grasping data (visual-tactile data) of 106 objects. It is collected with a sawyer hand mount with a different type of Gelsight sensors. Moreover, the RGB image is captured from a third-person view.

- **Taxim Synthetic Dataset (Taxim)** (Si & Yuan, 2022). We generate a synthetic dataset with Taxim's simulation and its tactile rendering. For RGB images, we use the same ray-tracing rendering. The dataset is curated with the same set of objects, physical properties, textures, backgrounds, and grasp poses. In addition, the dataset is sampled with the same method for a balanced distribution as our synthetic dataset. It contains 11k+ unique grasps and 33k+ visual-tactile data pairs. Please refer to the Appendix for details.

Instead of directly using raw tactile readings as input, we use the change of tactile reading, that is, tactile reading at grasp (with contact) minus tactile reading at initialization (without contact), similar to previous work (Calandra et al., 2018). For tactile inputs (2 $240 \times 320$ tactile images), they are encoded with a 4-layer CNN with BatchNorm and ReLU activation and then projected to two 512-dim embeddings. The $240 \times 320$ visual input is encoded with a separate CNN of the same architecture. We then concatenate all three embeddings and predict the binary output with a 2-layer MLP and Sigmoid output function. The model is trained with Adam (Kingma & Ba, 2017) optimizer with a learning rate of $3 \times 10^{-4}$ and a linear scheduler. In our evaluation, we run five seeds with the same configuration and select the checkpoint based on the validation accuracy for each run. We report the final result with the 5 checkpoints' accuracy on the testing dataset.

We find that models trained with the Real and the Taxim data have low performance (Figure 6). For the Real dataset, we believe this results from the large distribution shift. Although the data are collected under real-world physics and lightning, which have no sim-to-real gap, the data set is limited to the embodiment of the gripper, the sensor type, the camera angle, the objects and the environment. Moreover, it is very time-consuming to scale this dataset to open-world environments, larger sets of objects and grasps, and to different types of vision-based tactile sensors. Thus, it is not a scalable method for the open-world grasp stability prediction problem. For the Taxim dataset, we believe that it results from the large sim-to-real gap of the rigid body simulator and low-quality tactile rendering. Taxim (Si & Yuan, 2022) has only evaluated grasps of the same object, but not on open-world tasks, in which geometry, mass, will differ.

In contrast, our data generation method, based on the Taccel simulator, produces a model that achieves an average accuracy of 77.5% on the real-world testing data. This is a significant improve-

ment over the baselines. This highlights that our simulation data are a better training distribution for the open-world grasp stability prediction problem.

We also train models that use only vision inputs or with only tactile inputs from our dataset. We find that training with only tactile data has an average accuracy of 61.1% and training with only visual data comes from 53.2%, which is almost equivalent to a random guess. We hypothesize that using visual input alone is very challenging to predict the accurate grasp outcome in our dataset. Since we use an eye-in-hand camera, it is usually the case that half of the object is occluded (Figure 5). Thus, it is hard to infer the geometry or physical properties to predict the final result from RGB input alone. Consequently, tactile information becomes particularly important in our dataset. Not only does it reflect the normal contact surface of both fingers, but it also reflects the force of the grasp.

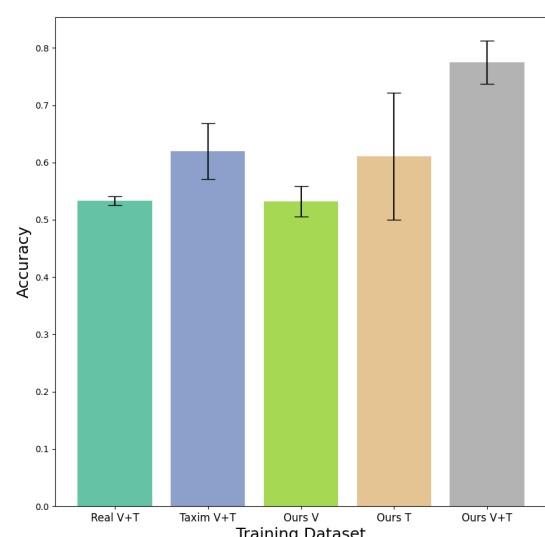

Figure 6: Results of model evaluation that are trained on different dataset. We show the mean and one standard deviation of the model's accuracy on the testing dataset, across 5 runs. Real V+T: Real dataset w/ vision+tactile inputs. Taxim V+T: Taxim dataset w/ vision+tactile inputs. Ours V: ours dataset w/ only vision inputs. Ours T: ours dataset w/ only tactile inputs. Ours V+T: ours dataset w/ vision+tactile inputs.

Although using tactile inputs alone achieves a relatively high performance, we find that it has large variance between runs, suggesting that using tactile information alone is not robust. Our best result uses both visual and tactile inputs. We hypothesize that understanding both contact information and partial information related to the object's geometry / physical properties (e.g., object's category and object's texture can provide information of the mass and friction) is the best recipe for the grasp stability prediction problem. A similar result has been discussed in previous work (Calandra et al., 2018).

## 6 CONCLUSION

We focus on learning open-world visual-tactile grasp stability predictor that can zero-shot generalize to novel objects in novel environments. Our work proposes training the model with synthetic data that is generated with FEM-based simulation and ray-tracing rendering. To this end, we have developed a synthetic data generator based on Taccel (Li et al., 2025b) and IsaacSim. With our pipeline, we generate a large-scale synthetic dataset of 30k+ visual-tactile data pairs and grasp outcomes.

To evaluate the prediction of zero-shot grasp stability, we collected an open-world dataset and a dataset with grasp pose annotations. In comparison to existing synthetic data generators like the Taxim (Si & Yuan, 2022), which is built-in rigid-body simulation, we show that our simulation pipeline has less sim-to-real gap and we produce more realistic tactile renderings. Moreover, predictors trained with our synthetic dataset have much higher real-world evaluation accuracy, compared to training with existing real datasets or synthetic data from the Taxim.

To learn better predictors, future work should include simulation of grasping deformable objects and scale the synthetic dataset with more diverse objects, environments, and grippers. It is also important to collect more real-world data for a more complete evaluation of different robots and applications.

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

# A  APPENDIX

## A.1  OBJECTS PREPARATION

**Mass and Friction Coefficient.** We use the following prompt to query the GPT-4o for estimates of the mass and friction coefficients. Here, *obj* denotes the object's category together with its texture / material, and *dim* is the largest dimension of its oriented bounding box.

```
Hi GPT, you are an expert in estimating mass and friction of
everyday objects. Please help me to estimate the mass of **{obj}**
of roughly {dim:.2f}m in size and its friction coefficient
when contacting with a rubber gel under dry condition.
In a new line, output the result in JSON format where the keys
are "mass" in kg and "friction" within [0.1, 1.5]. For example,
```{ "mass": 0.1, "friction": 1.2}.```
```

We find that in some cases the estimated mass is far from the range of testing objects. This is especially the case for object categories like Pianos, Cabinets, Desks that are not supposed to show on a table-top environment. While the ACRONYM dataset (Eppner et al., 2020) rescales them to a regular size for grasping, the LLM will estimate its corresponding mass based on normal size references in the real world. Thus, we limit the maximal mass of all objects to 1kg and then linearly map the mass of all objects to $[0.01, 0.2]$kg, which is the range of the real evaluation objects (Figure 3e).

**Texture.** Using GPT-4o (Achiam et al., 2023), we generate 5 textures for each object based on the label based on the following prompt.

```
Generate a list of 5 diverse textures / appearances modifiers
for {obj category}, no additional words, no numbers
```

We use the following prompt in Paint3D (Zeng et al., 2024) to generate the UV map for each object. We also add two additional stages of prompting to refine textures to be more realistic for each object.

```
UV map, {texture} {obj category}, photorealistic,
high quality, best quality
```

**Stats.** Figure 7 shows the statistics of the number of grasps of each object category in our synthetic dataset.

## A.2  TAXIM SIMULATION

**Physical Simulation.** Our adaptation of the Taxim (Si & Yuan, 2022) simulation framework employs a single parallel gripper as in Figure 8. Two GelSight Mini sensors are mounted on the gripper fingers. The entire system is simulated using PyBullet's rigid-body physics engine. The Taxim simulation places the object on the ground in its specified starting pose and the gripper at its specified grasping pose with an open width of 0.09 m. Then, the gripper closes with a specified grip force with a maximal of 150 steps. Visual-tactild data are recorded at this frame. The object is lifted 0.003 m per step for 60 steps, while maintaining gripping force. Finally, the pose of the object is recorded to for evaluation. We use the same procedure as in (Si & Yuan, 2022). Namely, if the object rises less than 80% of the expected lifting distance (0.144 m), the grasp is classified as a failure.

**Object Preparation.** Collision mesh of each object is computed with convex decomposition with PyBullet's V-HACD with $\alpha = 0.06$ and resolution $= 1e7$. We use the same mass, friction coefficient as in our synthetic dataset for the synthetic data generation with the Taxim.

**Tactile Rendering.** We adapted Taxim's default GelSight configuration to match our GelSight Mini sensor by adjusting the camera position and orientation for our mounting setup and recalibrating the LED positioning to reflect our sensor's internal lighting arrangement. Camera intrinsics and Taxim's original force-to-deformation mapping were preserved for consistency with the original simulation.

**Experiment Details in Section 5.1.** For each object, we run the simulation with friction coefficients of $\{0.1, 0.25, 0.4\}$, and grasping forces of $\{5, 10, 20\}$N. We first determine the optimal friction

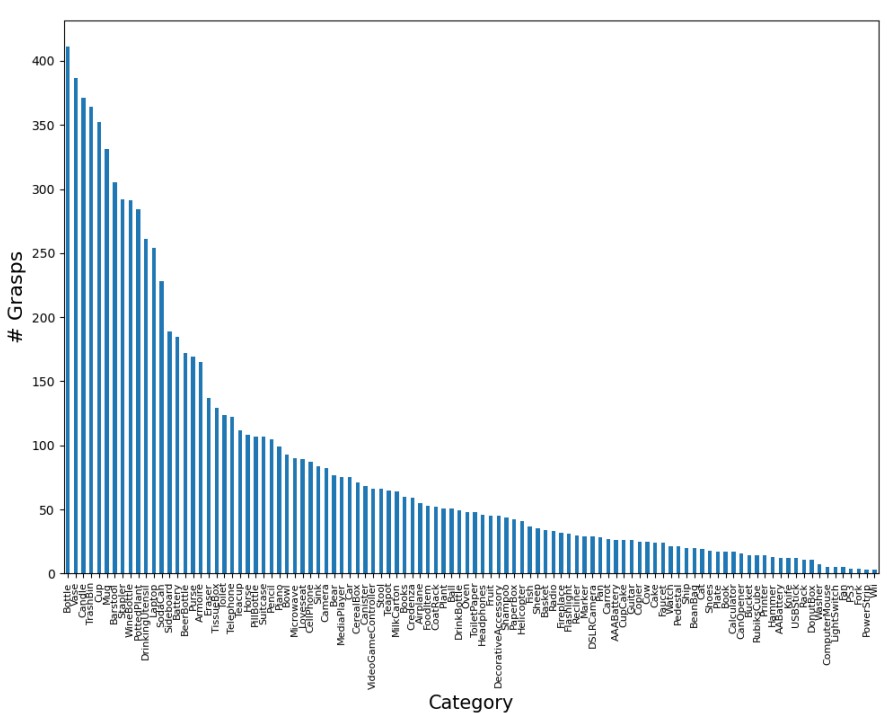

Figure 7: Number of grasps of each object category in our synthetic dataset.

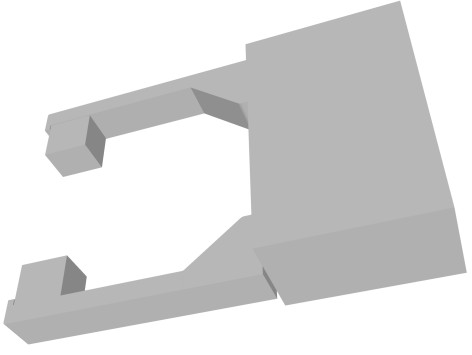

Figure 8: Gripper model with two GelSight Mini sensors used in Taxim simulation.

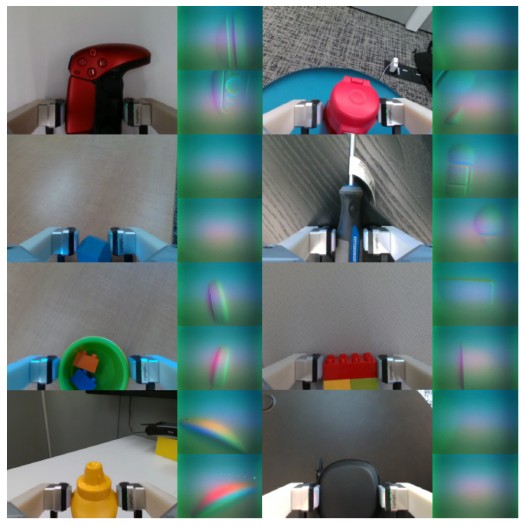 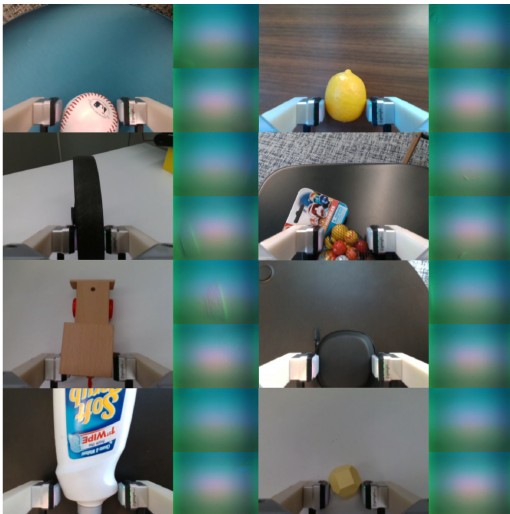

**Positive Grasps**          **Negative Grasps**

Figure 9: Examples of our open-world dataset. The left two columns show positive grasps and right two columns show negative grasps. Each case shows the RGB image, the left tactile reading (top) and the right tactile reading (bottom).

coefficient by selecting the value that minimizes the total number of incorrect predictions when using the best-performing force for each object. Them, we select the best gripping force for each object by comparing success/failure labels with ground truth results.

### A.3   OPEN-WORLD DATASET

We show more examples of our open-world dataset in Figure 9.

### A.4   GRASP-ANNOTATED DATASET AND EXPERIMENT DETAILS IN SECTION 5.1

We show the examples of our grasp-annotated dataset in Figure 10 for all 5 objects.

**Experiment Details.** Table 2 lists the configurations for each object. Here the mass is measured from real world. While friction coefficient is selected from $\{0.2, 0.4, 0.8\}$ and stopping force is selected from $\{20.0, 40.0, 80.0\}$N. We choose the same friction coefficient as the objects are printed with the same material.

Table 2: Simulation parameters in Section 5.1 of our simulator.

|  | B1 | B4 | C2 | C4 | ORG |
|---|---|---|---|---|---|
| Mass (g) | 70 | 65 | 57 | 65 | 43 |
| Friction Coefficient | 0.4 | 0.4 | 0.4 | 0.4 | 0.4 |
| Stopping Threshold $F_{\text{stop}}$ (N) | 80.0 | 40.0 | 40.0 | 20.0 | 40.0 |

### A.5   OUR SYNTHETIC DATASET

We show the examples of our synthetic training dataset in Figure 11.

### A.6   LLM USAGE

We use LLM to polish the language and modify the grammar. As discussed in A.1, we also use the LLM to prepare the objects for our dataset.

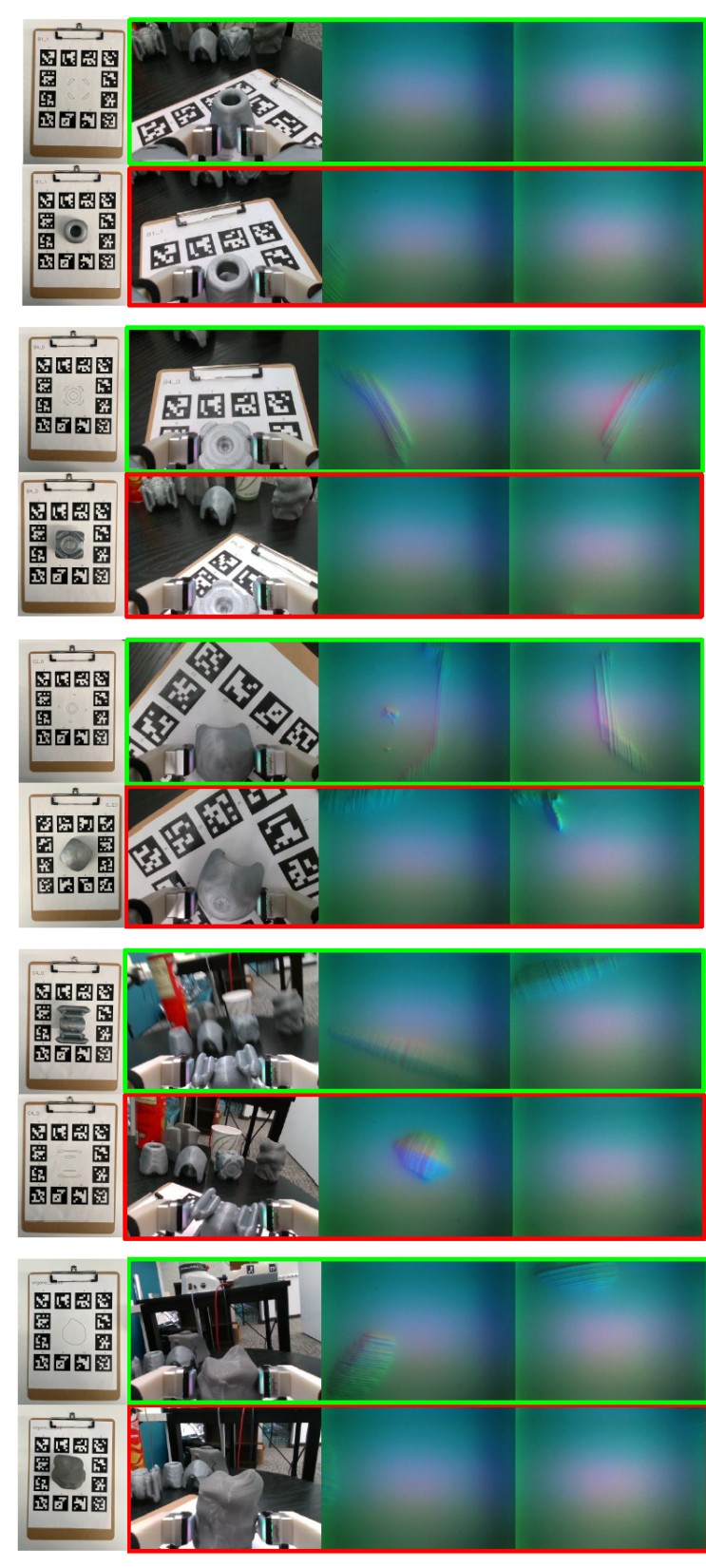

Figure 10: Examples of our grasp-annotated dataset. We demonstrate the contour line ArUco board on the left two images. We also include a positive grasp (top row) and a negative grasp (bottom row) data for each object.

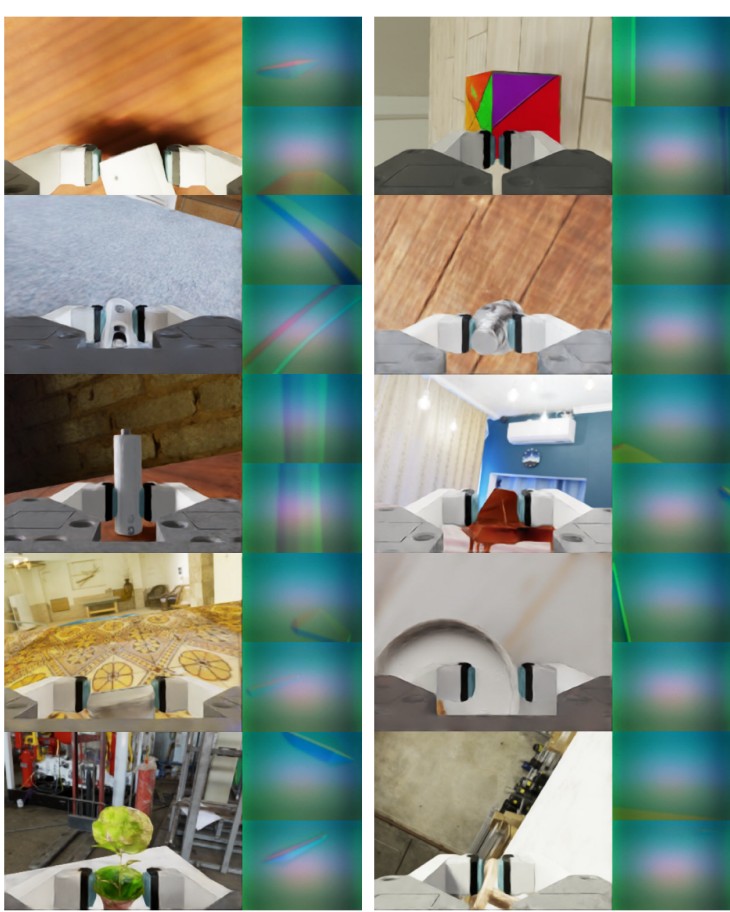

**Positive Grasps**          **Negative Grasps**

Figure 11: Examples of our synthetic training dataset. The left column shows the positive grasps and the right column shows the negative grasps.

