# OpenReview forum: "Learning Open-World Visual-Tactile Grasp Stability Prediction with Synthetic Data"
_ICLR.cc/2026/Conference — Submitted to ICLR 2026_

### Official Review · Reviewer_U1tn · 2025-11-01

**Soundness:** 2
**Presentation:** 2
**Contribution:** 2
**Rating:** 4
**Confidence:** 3

**Summary:**

This paper tackles open-world visual-tactile grasp stability prediction and proposes training entirely on synthetic data generated via FEM-based simulation (Taccel/IPC) plus ray-traced RGB and a custom tactile renderer. The pipeline produces ~30k visual-tactile pairs from 10k+ grasps across 453 objects, and the authors additionally collect a 333-grasp real-world “open-world” test set for zero-shot evaluation. A model trained on the synthetic set achieves 77.5% average accuracy on the real test set, outperforming training on an existing real dataset (9.2k samples) and on a rigid-body (Taxim/PyBullet) synthetic set. The paper also reports a smaller sim-to-real gap for FEM vs. rigid-body simulation on 5 grasp-annotated objects.

**Strengths:**

- Clear open-world setup with a purpose-built real evaluation set; synthetic data scale and diversity are well described (objects, scenes, lighting).
- Technically sound switch to FEM-based contact modeling and an improved tactile renderer; qualitative and quantitative evidence suggests reduced sim-to-real gap.
- Strong end-to-end gains: synthetic V+T training substantially beats real-only or rigid-body synthetic baselines; ablations show tactile is necessary and V+T is best.

**Weaknesses:**

- Real evaluation size is modest (266 test grasps) and human-actuated; potential bias across scenes/objects is under-analyzed (no per-scene/category/error breakdowns).
- Simulator fairness: in Sec. 5.1 you grid-search per-object friction $\mu_o$ and $F_\text{stop}$ to match outcomes, which can overfit the annotated set and handicap Taxim; cross-object fixed configs or cross-validation would be more convincing.
- Dataset preparation relies on LLM-estimated mass/friction and then clips/mass-maps to $[0.01,0.2]$ kg; this may distort contact dynamics. Reproducibility also hinges on releasing assets, renderer settings, and the 2500+ GPU-hour simulation outputs.

**Questions:**

- Tactile renderer: you intersect rays with the object (not the gel) and clip by gel thickness. Please provide the exact depth computation formula, LED positions/intensities, and any per-pixel normalization. How sensitive are results to these choices?
- Sec. 5.1 hyper-parameters: when tuning $\mu_o$ and $F_\text{stop}$ per object, which labels/frames were used, and how did you prevent test leakage? Can you report accuracy with (i) one global $(\mu_o,F_\text{stop})$ across all objects and (ii) parameters calibrated on a subset only?
- Simulation ablations: report performance sensitivity to $\Delta t=0.01$, $\kappa=3\!\times\!10^6$, $\hat d=5\!\times\!10^{-4}$, $\epsilon_v=10^{-3}$, $\epsilon_r=5\!\times\!10^{-5}$, and to the mass mapping/clipping. Also, will code, datasets (synthetic + real), and exact random seeds be released?

---

> ### Author Response · Authors · 2025-12-02
> **Response**
>
> 1.	**When tuning $\mu_o$ and $F_{stop}$ per object, which labels/frames were used, and how did you prevent test leakage? Can you report accuracy with (i) one global  across all objects and (ii) parameters calibrated on a subset only?**
>
> We use the label of the grasp outcome in the real-world. We show the parameters in Table 2. Here we set $\mu_o = 0.4$ to be the same as they are printed with the same material. We choose different $F_{stop}$, as this varies between different grasps / objects. However, we set them to the same in all 20 grasps of each object. We find that $F_{stop} = 40$ provides the good result except for B1 and C4. If we choose the same ($\mu_o = 0.4, F_{stop} = 40$), we get B1 with success rate 0.8 and C4 with success rate 0.85, that are still much higher than Taxim results.
>
> 2.	**Please provide the exact depth computation formula, LED positions/intensities, and any per-pixel normalization.**
>
> Given the depth map $D$, i.e., the distance from the camera plane to the first point on the object mesh (inf if no intersection). The camera plane is 0.02m back from the surface of the gel. Then, we compute the clipped $D$ = np.clip(D, 0.016, 0.02). Given the clipped depth map, we render the tactile reading with phone shading model [1]. Here the RED light is at [0, -1, -0.144] with RGB color [100, 0, 0], GREEN light is at [-1, 0, -0.144] with RGB color [0, 155, 0], and BLUE light is at [0, -1, -0.144] with RGB color at [0, 1, -0.144] with RGB color [0, 0, 155]. We choose $kd = 0.6$ and $ks = 0.1$ in the rendering model. There is no per-pixel normalization.
>
> 3.	**How sensitive is the performance to rendering parameters and gel’s physical parameters?**
> We find that using $\Delta t = 0.01$ and small error tolerance $\epsilon_r$ and $\epsilon_v$ are important for stable simulation of the IPC-framework [1], which is also mentioned in the Taccel paper. We choose the poisson ratio and young’s module as the rendering parameters as the same as in [2]. We find that parameters work well in our experiment without tuning.
>
> 4.	**Will code, datasets (synthetic + real), and exact random seeds be released?** Yes, we will release code, datasets, and model training script for reproducibility.
>
> [1] Li Y, Du W, Yu C, Li P, Zhao Z, Liu T, Jiang C, Zhu Y, Huang S. Taccel: Scaling Up Vision-based Tactile Robotics via High-performance GPU Simulation. arXiv preprint arXiv:2504.12908. 2025 Apr 17.
> [2] Li C, Dang R, Li X, Wu Z, Xu J, Kasaei H, Calandra R, Lepora N, Luo S, Su H, Chen R. ManiSkill-ViTac 2025: Challenge on Manipulation Skill Learning With Vision and Tactile Sensing. arXiv preprint arXiv:2411.12503. 2024 Nov 19.

---

### Official Review · Reviewer_eUvx · 2025-11-01

**Soundness:** 1
**Presentation:** 2
**Contribution:** 1
**Rating:** 2
**Confidence:** 4

**Summary:**

This paper proposed a simulation pipeline to generate synthetic dataset for training visual-tactile grasp stability predictors. The contributions are mainly based on a previous work Taccel for simulation. The authors claim that the trained grasp stability predictor on the synthetic dataset has higher prediction accuracy than baselines.

**Strengths:**

This paper collected a relatively large number of objects in the open-world dataset.

**Weaknesses:**

- This paper builds primarily on a previous work Taccel. The contributions appear to be mainly incremental and focused on engineering refinements, which somewhat limits the novelty and significance of the work.

- The paper would benefit from citing and discussing several relevant previous work, particularly to clarify how it advances beyond existing approaches. Please find more details below.

- The experimental results presented are somewhat limited in scope. Providing more comprehensive evaluations or additional analyses would make the contributions more convincing. Please find more details below.

- The authors mention grasping deformable objects as future work. However, using FEM-based simulation for rigid objects seems like mathematically and computationally overkill. It would be better to discuss more on this point.

**Questions:**

- The authors claim one of their contributions is to train visual-tactile models for grasp stability prediction but discussion and comparison of relevant work is missing, such as paper [1,2].

- In the related work section, several tactile sensor simulators are mentioned, but only Taxim is compared against the proposed pipeline in the experiments. It would be more convincing to compare with more simulators.

- The experiments focus exclusively on the GelSight Mini tactile sensor. The impact and generality of the work could be enhanced by considering other widely used tactile sensors, such as curved tactile sensor GelSight360.

- The evaluation currently covers a limited number of scenarios. Including a broader range of test cases or environments would provide a more comprehensive assessment.

References

[1] Cui, Shaowei, et al. "Self-attention based visual-tactile fusion learning for predicting grasp outcomes." IEEE Robotics and Automation Letters 5.4 (2020): 5827-5834.

[2] Cui, Shaowei, et al. "Grasp state assessment of deformable objects using visual-tactile fusion perception." 2020 IEEE International Conference on Robotics and Automation (ICRA). IEEE, 2020.

---

> ### Author Response · Authors · 2025-12-02
> **Response**
>
> 1.	**Discussion and comparison of relevant work is missing.**
>
> We thank the reviewer for referring related works that are not included. We will include them in the next version.
>
> 2.	**Only Taxim is compared against the proposed pipeline in the experiments.**
>
> To the best of our knowledge, Taxim is the only prior work that proposes a synthetic data generator for grasp stability prediction. Though simulation method for tactile readings has been proposed, they have not been developed and validated for our task.
>
> 3.	**The experiments focus exclusively on the GelSight Mini tactile sensor.**
>
> Yes. We follow the experiment setting used in prior works [1], [2]. Sensors like Gelsight360 is primarily used in Allegro Dexterous Hand and we leave it in future work.
>
> 4.	**The evaluation currently covers a limited number of scenarios.**
>
> Our evaluation dataset is wide. We collect 333 grasps on 60 objects in 10 scenes. We will keep expanding the dataset in the future.
>
> [1] Si Z, Yuan W. Taxim: An example-based simulation model for gelsight tactile sensors. IEEE Robotics and Automation Letters. 2022 Jan 13;7(2):2361-8.
> [2] Calandra R, Owens A, Upadhyaya M, Yuan W, Lin J, Adelson EH, Levine S. The feeling of success: Does touch sensing help predict grasp outcomes?. arXiv preprint arXiv:1710.05512. 2017 Oct 16.

---

### Official Review · Reviewer_HQ9U · 2025-11-01

**Soundness:** 3
**Presentation:** 3
**Contribution:** 3
**Rating:** 6
**Confidence:** 4

**Summary:**

This paper proposes a custom tactile simulator and renderer for large-scale synthetic data generation to train a grasping stability predictor. The authors demonstrate that the predictor generalizes to real-world and novel objects.

**Strengths:**

* The extension to diverse-object grasping prediction is strong and well supported.
* The authors describe the experimental settings in detail, making the work easier to validate and replicate.
* The ablation study shown in Fig 6 demonstrates the effectiveness of vision+tactile sensing compared to baselines.

**Weaknesses:**

* The proposed simulated rendering is better than the baseline, but it still appears quite different from real tactile patterns. According to Figs. 2 and 4, the simulated rendering looks bulky compared with fine-grained real tactile signals. Additionally, the simulated contact patterns differ: in Fig. 4 (right column), the real tactile image shows one line of contact points, whereas the simulated image shows two rows. This raises concerns about the effectiveness of the simulated tactile rendering. If the selected results differ so much from real observations, how can we be confident the stability predictor will produce useful predictions?
* The authors attribute the failure of Taxim to convex decomposition, but this argument is not fully convincing. The fidelity of convex decomposition can be tuned; if slower simulation is acceptable, artifacts can, in principle, be mitigated.
* Some contributions are not well explained. The paper appears to emphasize two main contributions: (1) novel-object grasping prediction and (2) a custom tactile simulation renderer. However, the renderer is not sufficiently elaborated and reads as an engineering addition on top of Taccel. Moreover, its quality does not align well with real measurements, as noted above.

**Questions:**

* Fig. 2 seems to indicate that the simulated tactile image differs substantially from the ground-truth tactile image. Is this reading correct? Additionally, the contact positions appear offset.
* How does the framework close the sim-to-real gap given the pronounced differences in visuo-tactile renderings? Directly evaluating the predictor in the real world may be brittle under such a gap.
* A suggestion: include three bullet points summarizing the paper’s contributions at the end of the introduction. This would make the paper clearer and aligns with common practice in the robotics community.

---

> ### Author Response · Authors · 2025-12-02
> **Response**
>
> 1.	**If the selected results differ so much from real observations, how can we be confident the stability predictor will produce useful predictions? Additionally, the contact positions appear offset.**
>
> The difference between real-world and simulated readings are much smaller comparing to Taxim simulation. Though there is still a gap, the information contained in the simulated readings are sufficient to acquire a better grasp stability predictor than using other data source.
> Moreover, the gap of Fig. 4 right column primarily results from the imprecise grasp pose annotation. Here we replay the grasp poses (Fig. 3) that are subject to annotation error. In terms of the grasp contact points on the object, it is susceptible to millimeters of pose difference.
>
> 2.	**The authors attribute the failure of Taxim to convex decomposition, but this argument is not fully convincing.**
>
> The failure of Taxim primarily comes from the difference of rigid-rigid contact in Taxim (Pybullet) versus the real-world soft-rigid contact dynamics. It is also effected by inaccurate convex decomposition artifacts. In our experiment, we use a fine convex decomposition of the original mesh.
>
> 3.	**Some contributions are not well explained. The renderer is not sufficiently elaborated and reads as an engineering addition on top of Taccel.**
>
> Our work develops a synthetic simulation data generation pipeline for training open-world visual-tactile grasp stability predictor. We show that our data is a better training data source than existing simulation and real-world data. As part of the pipeline, we develop a different tactile renderer for higher-quality tactile rendering.

---

### Meta-Review · Area_Chair_zkfn · 2026-01-02

**Summary:**

This paper studies open-world visual–tactile grasp stability prediction and proposes training predictors entirely on synthetic data generated via FEM-based simulation and rendering. Reviewers acknowledge that the problem is meaningful and that the synthetic pipeline leads to measurable gains over existing rigid-body simulation and limited real-data baselines. However, the contribution is largely viewed as incremental, building on prior tactile simulation frameworks with engineering refinements, and the empirical evidence does not fully establish generality or robustness at the level expected for ICLR. Despite clarifications and added analysis in the rebuttal, the work remains below the ICLR bar. The AC recommends rejection.

**Reviewer Concerns:**

Reviewer concerns focus on the depth, generality, and rigor of the evaluation. While the FEM-based simulation and tactile rendering reduce the sim-to-real gap relative to rigid-body baselines, the approach is seen as an extension of existing simulation pipelines rather than a clear modeling or learning advance. Questions remain about the fidelity of the simulated tactile signals, the fairness of simulator comparisons, and the sensitivity to simulation and rendering parameters. Evaluation on real-world data is limited in scale, sensor diversity, and scenario coverage, with insufficient analysis of bias, failure cases, and generalization. Although the rebuttal provides clarifications and additional implementation details, these concerns remain largely unresolved.

**Reviewer Scores:**

- Reviewer HQ9U: Scores the paper 6, recognizing the relevance of the problem and solid experimental effort, but expressing reservations about simulation fidelity and contribution clarity. After the rebuttal, the assessment would remain borderline (6).
- Reviewer eUvx: Scores the paper 2, viewing the contribution as largely incremental and the evaluation limited. This assessment would remain unchanged (2).
- Reviewer U1tn: Scores the paper 4, acknowledging strengths in data scale and empirical gains but raising concerns about fairness, generality, and reproducibility. After the rebuttal, the assessment would remain below the acceptance threshold (4).

---

### Decision · Program_Chairs · 2026-01-26

Reject